# Human papillomavirus vaccine efficacy against invasive, HPV-positive cancers: population-based follow-up of a cluster-randomised trial

Matti Lehtinen [1,2] Camilla Lagheden,[1] Tapio Luostarinen [3] Tiina Eriksson,[4] Dan Apter,[5] Anne Bly,[2] Penelope Gray,[1] Katja Harjula,[2] Kaisa Heikkilä,[2] Mari Hokkanen,[2] Heidi Karttunen,[2] Marjo Kuortti,[6] Pekka Nieminen,[7] Mervi Nummela,[2] J Paavonen,[8] Johanna Palmroth,[9] Tiina Petäjä,[10] Eero Pukkala,[11] Anna Soderlund-Strand,[12] Ulla Veivo,[2] Joakim Dillner [13]

For numbered affiliations see end of article.

**Correspondence to**
Dr Matti Lehtinen;
inkeri.lehtinen@gmail.com

## ABSTRACT

**Background** Human papillomavirus (HPV) vaccination protects against HPV, a necessary risk factor for cervical cancer. We now report results from population-based follow-up of randomised cohorts that vaccination provides HPV-type-specific protection against invasive cancer.

**Methods** Individually and/or cluster randomised cohorts of HPV-vaccinated and non-vaccinated women were enrolled in 2002–2005. HPV vaccine cohorts comprised originally 16–17 year-old HPV 16/18-vaccinated PATRICIA (NCT00122681) and 012 trial (NCT00169494) participants (2465) and HPV6/11/16/18-vaccinated FUTURE II (NCT00092534) participants (866). Altogether, 3341 vaccines were followed by the Finnish Cancer Registry in the same way as 16 526 non-HPV-vaccinated controls. The control cohort stemmed from 15 665 originally 18–19 years-old women enrolled in 2003 (6499) or 2005 (9166) and 861 placebo recipients of the FUTURE II trial. The follow-up started 6 months after the clinical trials in 2007 and 2009 and ended in 2019. It was age aligned for the cohorts.

**Findings** During a follow-up time of up to 11 years, we identified 17 HPV-positive invasive cancer cases (14 cervical cancers, 1 vaginal cancer, 1 vulvar cancer and 1 tongue cancer) in the non-HPV-vaccinated cohorts and no cases in the HPV-vaccinated cohorts. HPV typing of diagnostic tumour blocks found HPV16 in nine cervical cancer cases, HPV18, HPV33 and HPV52 each in two cases and HPV45 in one cervical cancer case. The vaginal, vulvar and tongue cancer cases were, respectively, positive for HPV16, HPV52/66 and HPV213. Intention-to-treat vaccine efficacy against all HPV-positive cancers was 100% (95% CI 2 to 100, p<0.05).

**Interpretation** Vaccination is effective against invasive HPV-positive cancer.

**Trial registration number** NCT00122681, Post-results; NCT00169494, Post-results; NCT00092534, Post-results.

## Strenghts and limitations of this study

► Individually and/or age-cluster randomised cohorts of originally adolescent women received human papillomavirus (HPV) vaccination, control vaccine vaccination or no vaccination in 2002–2005.
► All study participants consented to unique personal identifier facilitated long-term follow-up by the population-based Finnish Cancer Registry.
► All study participants had a possibility to attend cervical screening at the age of 25 and 30 years.
► All diagnostic biopsies with cervical or other HPV-associated cancers were retrieved for HPV DNA typing.
► Our HPV-vaccinated cohorts stem from the two largest clinical phase III trials, with 10% and 25% Finnish involvement and maximum, age-aligned follow-up time of up to 15 years, but the sample sizes are limited.
► Opportunistic, postlicensure HPV vaccination was negligible, probably had no effect on cervical precancer incidence but could not be totally ruled out.

(ICC) overall is lower among HPV-vaccinated women as compared with non-HPV-vaccinated women.[4] From population-based cancer registry follow-up of randomised cohorts, we have previously reported a significant vaccine efficacy (VE) against all invasive HPV-associated cancers.[5] We have now extended the follow-up time to 11 years and performed HPV typing of diagnostic tumour blocks. Our objective is to report the first HPV-type-specific VE estimates against invasive cancers in a randomised setting.

## INTRODUCTION

Human papillomavirus (HPV) vaccines are effective against cervical intraepithelial neoplasia grade 3 (CIN3) 8–14 years postvaccination.[1–3] The risk of invasive cervical cancer

## METHODS
### Enrolment, intervention and follow-up

All the 22 412 Finnish women born in annual quartal (Q)4/1984-Q1/1987 were invited to

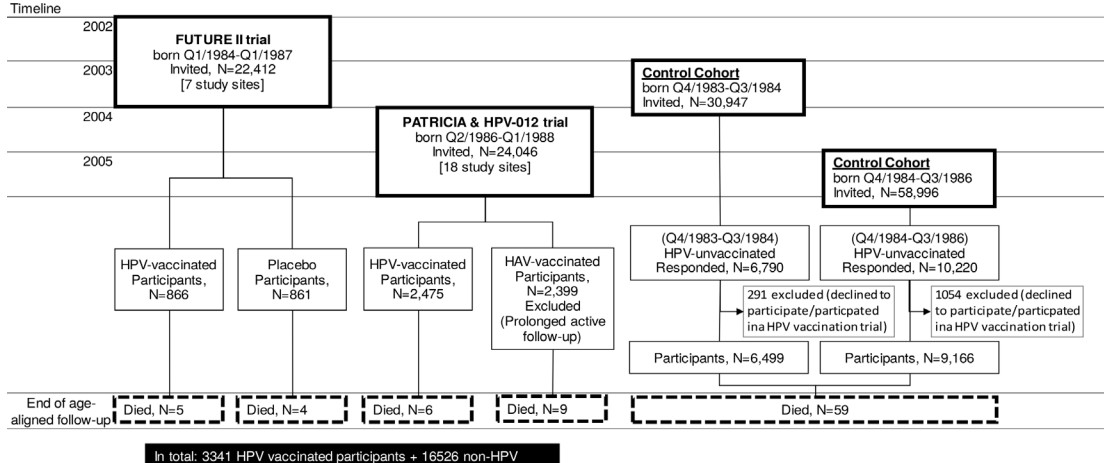

**Figure 1** Study flowchart defines the non-overlapping birth cohorts enrolled to long-term follow-up of human papillomavirus vaccination trials involving the quadrivalent HPV6/11/16/18 vaccine (FUTURE II) or the bivalent HPV16/18 vaccine (PATRICIA and HPV-012), or two non-HPV vaccinated control cohorts by population-based invitations via Finnish Population Register in 2002–2005. HPV, human papillomavirus; Q, annual quartal.

participate: (1) the FUTURE II (NCT00092534)[6] phase III HPV6/11/16/18 vaccine trial at age 16–17 years in seven towns in Q4/2002-Q1/2003. 24 064 women born Q2/1986-Q1/1988 were invited to participate; (2) the PATRICIA (NCT00122681)[7] phase III trial and (3) the HPV-012 (NCT00337818)[1] phase II trial also at the age 16–17 years in 18 Finnish towns in Q2/2004-Q1/2005 (figure 1). There were 3341 16–17 year-old HPV-vaccinated trial participants (by trial: (1) 866, (2) 2409 and (3) 66) (figure 1).[1 6 7]

In June 2003 and June 2005, all (30 947 and 58 996) non-HPV-vaccinated, 18 and 18–19-year-old Finnish women from the entire, immediately adjacent birth cohorts, born, respectively, in Q3/1984-Q2/1985 and Q3/1985-Q2/1987, not eligible to the above-mentioned clinical trials, were invited to participate in Finnish Cancer Registry (FCR)—based follow-up (figure 1).[6 7] Receiving HPV vaccination was an exclusion criterion. Respectively, in June 2003 and June 2005, 6499 and 9166 18–19-year-old non-HPV-vaccinated women were enrolled (figure 1).[6 7] During Q4/2002-Q1/2003 additional 861 16–17-year-old placebo recipients of the FUTURE II trial[7] were enrolled (figure 1).

All the clinical trial participants and the non-HPV-vaccinated controls consented to the FCR-based follow-up for invasive, histopathologically confirmed cancer outcome.[1 6–10] The follow-up was based on unique Finnish personal identifiers available at our study database following the informed consent and generally used at the FCR. It was age aligned[8] to ensure that all the originally 16–17-year-old HPV-vaccinated women and originally 16–19-year-old non-HPV-vaccinated women were of similar age during comparable follow-up time period of 11 years between 2007 and 2019.

All participants of the FCR-based follow-up were invited to cervical screening at the ages of 25, 30 and 35 years. For those with abnormal cytology during the trials, there was referral to colposcopy with biopsy within 6 months[1]

and the FCR follow-up, thus, started 6 months after the trials' end.

All the cohorts responded at the age of 22 years to a life style questionnaire, with special emphasis on sexual health and previous HPV vaccination. Death and emigration were elimination criteria. For the non-HPV vaccinated, any identified HPV vaccination was an elimination criterion. All the clinical HPV vaccination trials, the long-term follow-up of the trial cohorts and the non-HPV-vaccinated cohort were approved by the Finnish National Ethical Review Board (ERB, TUKIJA 1150/2002, 1153/2003, 1174/2004).

### Patient and public involvement

Patients were not involved in this preventive intervention study that involved originally adolescent HPV vaccinated or unvaccinated young women. To guarantee public involvement, the entire target population was duly informed by letters using the up to date address service of the Finnish Population Register. The trial invitations were mailed to parents or guardians of all adolescents girls of the above-mentioned birth cohorts and living in the same household. This was done even if the girl's own informed consent was sufficient for participation according to the Finnish law. Public involvement in the enrolment was further assured by information lectures given by the study nurses at parental evenings of the secondary high schools and technical schools in the 7 and 18 trial communities.

### Laboratory analyses

The diagnostic histopathological blocks were requested from the pathology laboratories that had notified the FCR about the incident cancer cases according to a specific permission from the Finnish National Supervisory Authority for Welfare and Health (Valvira) without informing the patients. Before HPV DNA typing, the presence of neoplastic tissue was rereviewed by two experienced pathologists. Blocks were sectioned according to

a contamination-proof manner. Extraction, amplification and typing of the intralesional HPV DNA were performed as previously described.[11 12] HPV types that remained unknown were identified by sequencing.

## Statistical analyses

The enrolled HPV-vaccinated and unvaccinated adolescents provided 80% power for the identification of VE against cervical cancer.[6 7] The VE was calculated on the intention-to-treat principle including all individuals (regardless of baseline HPV status) who had received at least one vaccine dose in the HPV vaccine arm. Statistical software SAS V.9.4 (SAS Institute, Cary, North Carolina) was used. The 95% CIs were based on exact binomial distribution of number of vaccinated cases conditional on total number of cases.[13]

## RESULTS

The age-aligned FCR follow-up for incident invasive cancers lasted by cohort for up to 11 years between November 2007 and December 2019. According to Finnish Population Census Register altogether, 10 FUTURE II, 19 PATRICIA and 73 non-HPV-vaccinated participants died during the follow-up (figure 1). It resulted in, respectively, 33 792 and 174 340 follow-up years for the HPV vaccine and non-HPV vaccine cohorts.

No HPV-associated cancer cases were found in the 3341 HPV-vaccinated women through the 33 792 women years of the follow-up (table 1). The follow-up was identified in the 16 526 non-HPV-vaccinated women 15 cases with ICC in the 16 526 non-HPV-vaccinated women during 174 320 women years of follow-up. Four of the ICC cases were adenocarcinomas and 11 were squamous cell carcinomas (SCC). One cervical SCC case, diagnosed late in 2019, was outside the age-aligned time-window of 11 years that guaranteed equal follow-up for the different cohorts. This case was not included in further analyses. The incidence of the 14 eligible ICC cases in the non-HPV-vaccinated women was 8.0 per 100 000 women years (table 1).

The single, invasive vaginal, vulvar and tongue cancer cases were all squamous cell carcinomas. Overall, in the non-HPV-vaccinated women the incidence of the HPV-associated cancers was 9.8 per 100 000 women years. Incidence rates of breast cancer, thyroid cancer and melanoma were essentially similar in all cohorts with highly overlapping 95% confidence intervals (table 1).

HPV DNA was found in all the identified and eligible 17 invasive HPV-associated cancer cases (table 2). HPV16 was present in nine ICC cases. HPV16 was present in one woman with vaginal cancer, who also had HPV16 positive cervical cancer. HPV18, HPV33 and HPV52 were each present in two ICC cases, and HPV45 in one ICC case. HPV66 (a vulvar cancer also positive for HPV52) and HPV213 (a tongue cancer) were present in one case each (table 2). With no identified HPV-associated cancer cases in the HPV-vaccinated women, we found a VE of 100% (95% CI 2 to 100; p<0.05) against all HPV DNA-positive invasive cancers.

## DISCUSSION

We report the first randomised trial-derived evidence on the significant, 100% HPV-VE against HPV DNA-positive invasive cancers. We followed both the bivalent and the quadrivalent HPV vaccine recipients for 11 years after the clinical trials had ended (up to 17 years postvaccination), and similarly aged non-HPV-vaccinated women, respectively, over approximately 35 000 and 175 000 follow-up years by the population-based, quality-controlled FCR.[9 10] VE estimates against invasive HPV16/18 positive cancers and against invasive cancer cases positive for the bivalent vaccine cross-protected HPV types 33/45/52[14–16] were all 100%, although with very wide CIs.

The significant 100% VE against all HPV-positive cancers is reassuring for both the bivalent and quadrivalent HPV vaccines, three shots of which were given to, respectively, 26% and 74% of the vaccinees in 2002 and 2004. A recent Swedish study,[4] which was also based on accurate, high-quality cancer registry follow-up,[13] suggested that up to

**Table 1** Numbers (n) and incidence rates (/100 000 person years) of human papillomavirus (HPV) associated invasive cancers and other common cancers in cluster-randomised cohorts of altogether 3341 16–17 year-old female HPV16/18 or HPV6/11/16/18 vaccine recipients and 16 526 non-HPV vaccinated, originally 16–19 year-old females followed up* between 2007 and 2019

| End-point | HPV-vaccinated women (33 792 person years) | | Non-HPV-vaccinated women (174 340 person years) | |
|---|---|---|---|---|
| | n | Rate (95% CI) | n | Rate (95% CI) |
| Cervical cancer | 0 | – | 14 | 8.0 (4.8 to 13.6) |
| All HPV-positive cancers† | 0 | – | 17 | 9.8 (6.1 to 15.7) |
| Breast cancer | 3 | 8.9 (2.9 to 28) | 27 | 15.5 (10.6 to 23) |
| Thyroid cancer | 2 | 5.9 (1.5 to 24) | 16 | 9.2 (5.6 to 15.0) |
| Melanoma | 8 | 23.8 (11.8 to 47) | 22 | 12.6 (8.3 to 19.2) |

*For corresponding age-aligned sub-cohorts, up to 11 years of passive follow-up was by the population-based Finnish Cancer Registry.
†14 cervical cancers, one vaginal cancer, one vulvar cancer, one tongue cancer.

**Table 2** Numbers (n) and incidence rates (/100,000 person years) of human papillomavirus (HPV) positive invasive cancers by HPV type in cluster-randomised cohorts of altogether 3341 16–17 year-old female HPV16/18 or HPV6/11/16/18 vaccine recipients and 16 526 non-HPV vaccinated, originally 16–19 year-old females followed up* during 2007–2019

| End-point | HPV-vaccinated women (33 792 person years) | | Non-HPV-vaccinated women (174 340 person years) | |
|---|---|---|---|---|
| | n | Rate (95% CI) | n | Rate (95% CI) |
| Cervical cancer | | | | |
| HPV16 | 0 | – | 9 | 5.2 (2.7 to 9.9) |
| HPV18 | 0 | – | 2 | 1.1 (0.3 to 4.6) |
| HPV16/18 | 0 | – | 11 | 6.3 (3.5 to 11.4) |
| HPV33 | 0 | – | 2 | 1.1 (0.3 to 4.6) |
| HPV45 | 0 | – | 1 | 0.6 (0.1 to 4.1) |
| HPV52† | 0 | – | 1† | 0.6 (0.1 to 4.1) |
| Any HPV | 0 | – | 14 | 8.0 (4.8 to 13.6) |
| Vaginal cancer | | | | |
| HPV16 | 0 | – | 1 | 0.6 (0.1 to 4.1) |
| Vulvar cancer | | | | |
| HPV52‡ | 0 | – | 1 | 0.6 (0.1 to 4.1) |
| Tongue cancer | | | | |
| HPV213 | 0 | – | 1 | 0.6 (0.1 to 4.1) |

*For corresponding age-aligned sub-cohorts, up to 11 years of passive follow-up was by the population-based Finnish Cancer Registry.
†Positive for both HPV16 and HPV52.
‡Positive for both HPV52 and HPV66.

12% of ICC cases may not be prevented by the quadrivalent vaccine. We have recently shown that long-term cross-neutralising antibody responses induced by these two vaccines differ significantly.[16] Thus, continuation of the cancer registry-based long-term follow-up of the population-based bivalent and the quadrivalent HPV vaccine cohorts versus non-HPV-vaccinated cohorts[6 7] is warranted to understand if there were a possible difference in their efficacy.

Population-based, age-aligned follow-up of trial cohorts and accurate, histologically confirmed, invasive cancer case definitions of the quality-controlled FCR[6–8 12 13 17] were pivotal strengths of this study. We could ensure identical cervical screening history of HPV-vaccinated and non-HPV-vaccinated cohorts for the 1983–1988 birth cohorts that were included in this study. The 5-year interval screening at ages 25, 30, 35… is according to local standard of care.

During 2007–2009, entire 1992–1995 birth cohorts of 33 Finnish towns were enrolled in our population-based community-randomised HPV vaccination trial.[18 19] HPV16/18 vaccine recipients of that trial have since 2014 participated in a rerandomised screening trial at ages 22, 25 and 28 years,[20] while their non-HPV-vaccinated counterparts have not had this opportunity. These sizeable cohorts, >40 000 individuals in total, were not included in this study due to their gradually more and more different screening histories.

Two independent pathologists ensured that neoplastic tissue was present in the sections of the initial diagnostic biopsy blocks that we wanted to use for HPV testing. In three ICC cases, CIN3 was left in the diagnostic block that was used for HPV DNA typing, but the sectioned material was deemed to adequately represent oncogenic HPV types involved. The blocks were found to be positive for HPV16, 33 and 16/52 DNA. Conceivably, due to the lesion size in the diagnostic block, prior sections had removed an ICC before the new sections were made for the HPV typing.

Recruiting and enrolling the different cohorts in a population-based fashion during 2002–2005 was an important prerequisite for the country-wide cancer registry-based follow-up.[6–8 17] Due to the follow-up evidence on protection against the most stringent, invasive HPV-associated cancer end points are now emerging more rapidly[4 5] than the evidence on the efficacy of HBV vaccination against hepatocellular cancer.[21] HPV cancers in young adults may be more rapidly developing but establishing the trial cohorts and concomitant control cohorts, and their follow-up 17 years ago with identical possibilities for organised cervical screening at ages 25, 30 and 35 years[1 17] has been a strength of this study. At the time of HPV vaccine licensure (2006/2007), the control cohort was above 20 years of age, and opportunistic vaccination was negligible but could not be totally ruled out,[22 23] which should be listed as a limitation of this study. On the other hand, cross-vaccinating 50% of the FUTURE II placebo arm after age 20 had no material effect on their CIN3 incidence.[8]

The excellent VE against HPV-positive cancers now documented from a randomised study setting, it is an important evidence of the long-term impact of HPV vaccination. It indicates that prevention of a sexually transmitted infection and associated cancer by prophylactic vaccination are doable and paves the way for the WHO's initiative on the elimination of cervical cancer.[24]

**Author affiliations**
¹Department of Laboratory Medicine, Karolinska Institutet, Stockholm, Sweden
²Tampere University Hospital, Tampere, Finland
³Finnish Cancer Registry, Helsinki, Finland
⁴FICAN-Mid, Tampere, Finland
⁵VL Medi, Helsinki, Finland
⁶Faculty of Social Sciences, Tampereen Yliopisto, Tampere, Finland
⁷Helsingin Yliopisto, Helsinki, Finland
⁸Department of Gynecology and Obstetrics, University of Helsinki, Helsinki, Finland
⁹University of Eastern Finland School of Medicine, Kuopio, Pohjois-Savo, Finland
¹⁰Obstetrics & Gynecology, Tampereen Yliopisto, Seinäjoki, Finland
¹¹Cancer Society of Finland, Helsinki, Finland
¹²Clinical Microbiology, Lund University, Lund, Sweden
¹³Department of Medical Epidemiology and Biostatistics, Karolinska Institutet, Stockholm, Sweden

**Acknowledgements** The review of diagnostic histopathological blocks for HPV DNA analysis by Drs Mensur Dzabic and Ralf Butzow is gratefully acknowledged.

**Contributors** The corresponding author was the guarantor for the overall content and attests that all listed authors meet authorship criteria and that no others meeting the criteria have been omitted. ML, TL, DA, JD, TE, JPaa and EP were involved in the conception and/or the design of the study. ML, DA, AB, TE, KHa, KHe, MH, HK, MK, TL, PN, MN, JPal, TP, AS-S and UV participated in the collection and generation of the study data. TE, PG, CL and TL performed the analyses. ML, DA, TL, JPaa and EP were involved in the analyses and/or interpretation of the data. All authors contributed to development of this manuscript, had full access to the data and gave final approval before submission.

**Funding** Finnish Cancer Society, Swedish Cancer Society, Nordic Cancer Union, and GlaxoSmithKline Biologicals SA (study identifier 115006) supported the study, which discloses in the publication all the data on invasive cancers collected. GlaxoSmithKline Biologicals SA was provided the opportunity to review a preliminary version of this manuscript for factual accuracy, but the authors are solely responsible for final content and interpretation. GSK Biologicals SA (HPV-027, 115006) and Finnish Cancer Foundation (MS790) supported the study financially,

**Competing interests** We declare that ML, DA, JD and JPaa have received grants from Merck & Co. Inc. and/or from the GSK group of companies through their respective employers. GSK Biologicals SA was provided the opportunity to review this manuscript but the authors are solely responsible for final content and interpretation.

**Patient and public involvement** Patients and/or the public were not involved in the design, or conduct, or reporting, or dissemination plans of this research.

**Patient consent for publication** Not applicable.

**Provenance and peer review** Not commissioned; externally peer reviewed.

**Data availability statement** Data are available upon reasonable request. Data is available upon reasonable request from corresponding author.

**ORCID iDs**
Matti Lehtinen http://orcid.org/0000-0002-9481-0535
Tapio Luostarinen http://orcid.org/0000-0003-3231-8550
Joakim Dillner http://orcid.org/0000-0001-8588-6506

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
