## [Reviewer comments · BMJ Open]

ARTICLE DETAILS

TITLE (PROVISIONAL)	Human papillomavirus vaccine efficacy against invasive, HPV-positive cancers – Population-based follow-up of a cluster-randomized trial
AUTHORS	Lehtinen, Matti; Lagheden, Camilla; Luostarinen, Tapio; Eriksson, Tiina; Apter, Dan; Bly, Anne; Gray, Penelope; Harjula, Katja; Heikkilä, Kaisa; Hokkanen, Mari; Karttunen, Heidi; Kuortti, Marjo; Nieminen, Pekka; Nummela, Mervi; Paavonen, J; Palmroth, Johanna; Petäjä, Tiina; Pukkala, Eero; Soderlund-Strand, Anna; Veivo, Ulla; Dillner, Joakim

VERSION 1 – REVIEW

REVIEWER	Nicolaidou, Electra National and Kapodistrian University of Athens
REVIEW RETURNED	28-Jun-2021

GENERAL COMMENTS	This is a very interesting and very well written study on the protection offered by HPV vaccination against invasive cancer. The results of the study were highly anticipated and we are very happy to have them. I understand that the study was conducted by major and well-respected centers in the field. However, the average reader (like me) may need a few more clarifications regarding the study design. For example, I had to read the "Methods" section several times to understand the composition of the control group. So, my comments are as follows: 1) The "Enrollment, intervention and follow-up" part of the "Methods" is hard to follow. A bulleted text or a table may help.2) It was not clear to me why the non-vaccinated cohort did not receive the vaccine. Did they have the chance to receive the vaccine and they refused it or vaccination was never an option? In any case, the authors may want to comment on that in the Discussion.3) "All participants of the FCR-based follow-up were invited to cervical screening at the ages of 25, 30 and 35 years. For those with abnormal cytology during the trials, there was referral to colposcopy with biopsy within 6 months" The authors may want to comment on whether cervical screening every 5 years starting at age 25 and whether waiting for colposcopy for 6 months after abnormal cytology are in accordance with the national guidelines for cervical cancer screening.
--

REVIEWER	Della Corte, Luigi University of Naples Federico II
REVIEW RETURNED	03-Aug-2021

GENERAL COMMENTS	I was pleased to review this paper “Human papillomavirus vaccination protects against invasive HPV-positive cancers” by BMJ Open. 1. The methodology used by the Authors is appropriate for the study and conclusions are narrowly linked to data discussion and available evidence. The English language is fluid and well understood. Nevertheless, the paper is of good quality, I would have added just more recent studies, for this is need a minor revision. I suggest this: 1. Abstract: good. 2. Introduction: In a historical period such as this, it is essential to deal with articles of this type that inform the discussion on the function of vaccines. There is a good extended about the follow-up time to 11 years and performed HPV-typing of diagnostic tumor blocks. 3. Material and Methods. The first thing I would like to point out is the good number of the statistical sample and the functional study time. The statistics used are good and the mathematical interpretation given is correct. 4. Discussion. This paper reports the 100% HPV-vaccine efficacy against HPV DNA-positive invasive cancers is reassuring for both the bivalent and quadrivalent HPV vaccines.  • You could attach a table to define the familiar cases of expression of deer tumors. • You could also attach the questions administered on the sexual habits of the patients to verify the role of promiscuity in HPV infections. • You could read these articles to improve your understanding of the topic: doi: 10.1080/14656566.2020.1724284; doi: 10.1111/jog.14276. 5. Results and Conclusion. The results are consistent with the study. Corrected the method used in the work, given the statistical results obtained, in fact about non-HPV-vaccinated women there are cases of growth of cancer. This paper draws a fair conclusion concerning the work and international literature.
--

VERSION 1 – AUTHOR RESPONSE

Reviewer: 1

Dr. Electra Nicolaidou, National and Kapodistrian University of Athens

Comments to the Author:

This is a very interesting and very well written study on the protection offered by HPV vaccination against invasive cancer. The results of the study were highly anticipated and we are very happy to have them.

I understand that the study was conducted by major and well-respected centers in the field. However, the average reader (like me) may need a few more clarifications regarding the study design. For

example, I had to read the "Methods" section several times to understand the composition of the control group. So, my comments are as follows:

1) The "Enrollment, intervention and follow-up" part of the "Methods" is hard to follow. A bulleted text or a table may help.

We have improved the clarity of the Enrolment, Intervention and Follow-up chapter in the Methods section with special reference to the flowchart (figure 1).

2) It was not clear to me why the non-vaccinated cohort did not receive the vaccine. Did they have the chance to receive the vaccine and they refused it or vaccination was never an option? In any case, the authors may want to comment on that in the Discussion.

As now discussed in the end of page 11 to the vast majority of the controls only opportunistic vaccination after age 20 was possible albeit negligible. Furthermore, this had no effect on the CIN3 incidence of the cross-vaccinated young adult women as also discussed and duly cited with on page 11.

3)"All participants of the FCR-based follow-up were invited to cervical screening at the ages of 25, 30 and 35 years. For those with abnormal cytology during the trials, there was referral to colposcopy with biopsy within 6 months" The authors may want to comment on whether cervical screening every 5 years starting at age 25 and whether waiting for colposcopy for 6 months after abnormal cytology are in accordance with the national guidelines for cervical cancer screening..

Noted in the end of page 10 of the Discussion

Reviewer: 2

Dr. Luigi Della Corte, University of Naples Federico II

Comments to the Author:

I was pleased to review this paper "Human papillomavirus vaccination protects against invasive HPV-positive cancers" by BMJ Open.

1. The methodology used by the Authors is appropriate for the study and conclusions are narrowly linked to data discussion and available evidence. The English language is fluid and well understood. Nevertheless, the paper is of good quality, I would have added just more recent studies, for this is need a minor revision.

To the best of our knowledge pertinent recent papers on VE against the most stringent end-points are cited

I suggest this:

1. Abstract: good.

Thank you.

2. Introduction: In a historical period such as this, it is essential to deal with articles of this type that inform the discussion on the function of vaccines. There is a good extended about the follow-up time to 11 years and performed HPV-typing of diagnostic tumor blocks.

Using the diagnostic tumor blocks was pivotal but brought the problem with lesion size now discussed in paragraph 2 on page 11.

3. Material and Methods. The first thing I would like to point out is the good number of the statistical sample and the functional study time. The statistics used are good and the mathematical interpretation given is correct.

Thank you.

4. Discussion. This paper reports the 100% HPV-vaccine efficacy against HPV DNA-positive invasive cancers is reassuring for both the bivalent and quadrivalent HPV vaccines.

- You could attach a table to define the familiar cases of expression of deer tumors.

We regret but the cases are too few for the suggested tabulation.

- You could also attach the questions administered on the sexual habits of the patients to verify the role of promiscuity in HPV infections.
- You could read these articles to improve your understanding of the topic: doi: 10.1080/14656566.2020.1724284; doi: 10.1111/jog.14276.

Thank you for the point raised. We have revised the discussion on page 12 accordingly.

5. Results and Conclusion. The results are consistent with the study. Corrected the method used in the work, given the statistical results obtained, in fact about non-HPV-vaccinated women there are cases of growth of cancer. This paper draws a fair conclusion concerning the work and international literature.

Thank you. Please, see above for discussion on the pros and cons of using the diagnostic tumor blocks.

VERSION 2 – REVIEW

REVIEWER	Nicolaidou, Electra National and Kapodistrian University of Athens
REVIEW RETURNED	18-Oct-2021
GENERAL COMMENTS	From my end, the manuscript can now be accepted for publication.